# Insulin resistance markers HOMA-IR, TyG and TyG-BMI index in relation to heart failure risk: NHANES 2011-2016

**Meijuan Lu[1], Junchi Guo[2], Peili Yang[2], Teng Ma[3], Mingyan Zhang[4]***

**1** Department of Cardiology, The Second Affiliated Hospital of Tianjin University of Traditional Chinese Medicine, Tianjin, China, **2** Graduate School, Tianjin University of Traditional Chinese Medicine, Tianjin, China, **3** Department of Traditional Chinese Medicine, The First Affiliated Hospital of Zhengzhou University, Zhengzhou, China, **4** Center for Evidence-Based Medicine, Tianjin University of Traditional Chinese Medicine, Tianjin, China

◉ These authors contributed equally to this work.

* zmy157@126.com

## Abstract

### Background

Insulin resistance (IR) is increasingly recognized as an important factor in the development of heart failure (HF). This study aimed to evaluate the association and predictive ability of three IR markers—HOMA-IR, TyG, and TyG-BMI index—with HF risk.

### Methods

Data from 7,668 participants in the NHANES 2011–2016 survey were analyzed. Multivariable logistic regression was used to assess the relationship between HOMA-IR, TyG, and TyG-BMI with HF incidence, adjusting for potential confounders. Receiver operating characteristic (ROC) curves, decision curve analysis (DCA), and restricted cubic spline (RCS) analysis were conducted to compare the predictive performance of the three markers.

### Results

HOMA-IR (OR = 1.017, 95% CI: 1.006–1.027, P < 0.01), TyG (OR = 1.798, 95% CI: 1.453–2.225, P < 0.001), and TyG-BMI (OR = 1.006, 95% CI: 1.004–1.008, P < 0.001) were all significantly associated with HF risk, with TyG showing the strongest association. ROC curve analysis demonstrated that TyG (AUC = 0.61) and TyG-BMI (AUC = 0.62) had better predictive abilities than HOMA-IR (AUC = 0.6). In subgroup analyses, HOMA-IR showed higher sensitivity in the female population, while TyG-BMI provided a complementary role to TyG in individuals with diabetes.

**Data availability statement:** The data used in this study are publicly available from the National Health and Nutrition Examination Survey (NHANES), conducted by the Centers for Disease Control and Prevention (CDC), National Center for Health Statistics (NCHS). Data from the 2011–2016 cycles were included. The datasets used are accessible through the NHANES website (https://wwwn.cdc.gov/nchs/nhanes/), including: Demographics: DEMO_G (2011–2012), DEMO_H (2013–2014), DEMO_I (2015–2016); Laboratory: INS_G/H/I (Insulin), GLU_G/H/I (Fasting Glucose), TRIGLY_G/H/I (Triglycerides); Examination: BPX_G/H/I (Blood Pressure); Questionnaire: MCQ_G/H/I (Medical Conditions, including heart failure), DIQ_G/H/I (Diabetes Questionnaire). All datasets are publicly available without restrictions at the NHANES official repository. Additional data can be found in the Supporting information files submitted by the authors.

**Funding:** The authors declare that financial support was received for the research, writing, and/or publication of this article. This study was supported by the following grants: the National Natural Science Foundation of China General Project (No. 82374619), the National Natural Science Foundation of China Youth Project (No. 81904056). The funders had no role in study design, data collection and analysis, decision to publish, or preparation of the manuscript.

**Competing interests:** The authors have declared that no competing interests exist.

**Abbreviations:** IR, IR; HF, Heart failure; TyG, triglyceride glucose index; ROC, Receiver operating characteristic curves; DCA, Decision curve analysis; RCS, Restricted cubic spline analysis; AUC, Area under the curve; TC, Total cholesterol; LDL-C, Low-density lipoprotein cholesterol; HDL-C, High-density lipoprotein cholesterol; Na, Serum sodium; K, Serum potassium; UA, Uric acid; AST, Aspartate aminotransferase; ALT, Alanine aminotransferase; Cr, Creatinine; Alb, Albumin.

## Conclusion

TyG showed a stronger association with HF risk than HOMA-IR and TyG-BMI and outperformed HOMA-IR in predicting HF risk, particularly in specific subpopulations. These findings highlight the importance of further research into the clinical application of TyG for early identification and management of HF risk.

## 1. Introduction

Heart Failure(HF) is one of the leading cardiovascular diseases contributing to increasing morbidity and mortality worldwide, presenting a significant challenge to both patients' quality of life and healthcare systems [1,2]. Although numerous studies have identified various risk factors for HF, such as hypertension, diabetes, and coronary heart disease, growing evidence suggests that metabolic abnormalities, particularly insulin resistance(IR), play a critical role in its pathophysiology [3–5].

IR is a central component of metabolic syndrome, characterized by a reduced ability of insulin to regulate glucose and lipid metabolism [6,7]. It not only serves as a primary pathological basis for type 2 diabetes but is also closely linked to cardiovascular diseases [8]. Commonly used indicators for assessing IR include the Homeostasis Model Assessment of IR (HOMA-IR), the Triglyceride-Glucose Index (TyG), and the Triglyceride-Glucose Index and Body Mass Index Product (TyG-BMI). HOMA-IR is calculated using fasting insulin and fasting glucose levels and is widely employed in clinical and research settings [9]. However, its reliance on the standardization and accuracy of insulin assays limits its applicability in large-scale population studies. In contrast, TyG, which only requires fasting glucose and triglyceride levels, is more convenient, reliable, and has demonstrated strong performance in predicting diabetes and cardiovascular diseases [10]. Additionally, TyG-BMI, which combines the TyG index with BMI, may provide additional value in assessing IR and predicting cardiovascular disease by better reflecting an individual's fat distribution and metabolic status [11].

While studies have established significant associations between HOMA-IR, TyG and TyG-BMI with various cardiovascular disease risks, the differences between these three markers in predicting HF risk have not been thoroughly examined [12,13]. Moreover, the predictive performance of these IR markers may vary across subgroups defined by sex, age, and race. However, most existing studies focus on a single marker and fail to systematically compare the relative predictive strengths of HOMA-IR, TyG, and TyG-BMI for HF risk [14]. A systematic comparison of HOMA-IR, TyG and TyG-BMI across diverse populations could help identify the more appropriate IR markers for predicting HF risk, providing stronger evidence for clinical practice.

To address this gap, the present study systematically compares the associations between HOMA-IR, TyG, TyG-BMI and HF risk using data from NHANES 2011–2016. Subgroup analyses by sex, age, and race were conducted to explore the differences in the predictive power of these IR markers across different demographic groups. By conducting this stratified comparison, the study aims to provide a more

comprehensive understanding of the relative advantages of HOMA-IR, TyG and TyG-BMI index in predicting HF risk, and to identify the marker that may be more applicable to specific populations. This research provides a scientific basis for developing more targeted prevention and intervention strategies.

## 2. Materials and methods

### 2.1. Study population

This study utilized data from the 2011–2016 cycles of the National Health and Nutrition Examination Survey (NHANES). NHANES is a cross-sectional survey designed to evaluate the health and nutritional status of the civilian, non-institutionalized U.S. population [15,16]. The survey employs a complex, multistage probability sampling design to ensure national representativeness [17]. For this study, we applied the following inclusion and exclusion criteria: individuals aged 18–80 years with available data on insulin, fasting glucose, and triglycerides were included to calculate HOMA-IR, TyG and TyG-BMI index. Exclusion criteria were as follows: 1) age under 18 or over 80 years; 2) estimated glomerular filtration rate (eGFR) < 60 mL/min/1.73 m², indicating significant renal impairment; 3) pregnancy; and 4) missing or incomplete data on key variables, including insulin, fasting glucose, triglycerides, or HF diagnosis. This study used publicly available NHANES database data, and therefore, no additional ethical approval was required. A flowchart outlining the participant screening process is presented in Fig 1.

### 2.2. Assessment of HOMA-IR and TyG

In this study, three commonly used markers of IR were employed: HOMA-IR, TyG and TyG-BMI index. HOMA-IR is calculated using fasting insulin and fasting glucose levels, according to the formula: HOMA-IR = [fasting insulin (mU/L) × fasting glucose (mg/dL)]/ 405 [18]. HOMA-IR is widely used to assess IR, with higher values indicating greater degrees of resistance [19]. TyG is calculated using fasting triglycerides and fasting glucose, using the formula: TyG = ln [fasting triglycerides (mg/dL) × fasting glucose (mg/dL)/ 2] (13). The TyG index serves as an alternative marker for assessing IR and has shown strong correlations with IR in several studies [20]. In addition, the TyG-BMI is calculated by combining the TyG index with BMI, using the formula: TyG-BMI = TyG × BMI. TyG-BMI has been proposed as a novel marker for assessing IR, as it can integrate the influence of body weight factors on IR [21].

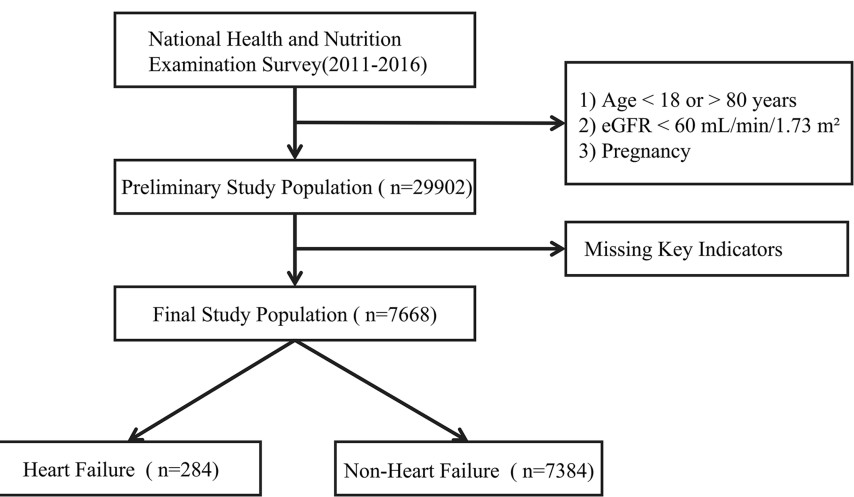

**Fig 1. Flowchart depicting the participant selection process in the study.**

Based on the distribution of HOMA-IR, TyG, and TyG-BMI values, participants were further categorized into four quartiles: Q1, Q2, Q3, and Q4, representing different ranges of HOMA-IR, TyG, or TyG-BMI values. Specifically, Q1 corresponds to the lowest quartile (HOMA-IR < X or TyG < A or TyG-BMI < B), while Q4 represents the highest quartile (HOMA-IR ≥ Y or TyG ≥ C or TyG-BMI ≥ D). These quartile groupings were used to explore the relationship between HOMA-IR, TyG, or TyG-BMI and the study outcomes.

## 2.3. Definition of HF

In this study, HF was defined based on a self-reported item (MCQ160b) from the NHANES questionnaire. The questionnaire was administered by specially trained interviewers, who asked participants whether a doctor or other health professional had ever informed them that they had HF. The specific question was: "Has a doctor or other health professional ever told you that you have HF?" Participants responded with either "yes" or "no" according to their actual condition. This self-reported data is widely utilized in epidemiological studies and is considered to have good reliability and validity in large-scale surveys.

## 2.4. Covariates

Multiple covariates were selected to control for potential confounding factors, including demographic variables and biochemical indicators. Sex (RIAGENDR) and age (RIDAGEYR) were obtained from the demographic questionnaire. Race/ethnicity (RIDRETH1) was categorized into five groups: non-Hispanic White, non-Hispanic Black, other Hispanic, Mexican American, and other racial groups. Biochemical covariates included low-density lipoprotein cholesterol (LBDLDLSI), high-density lipoprotein cholesterol (LBDHDDSI), serum sodium (LBXSNASI), serum potassium (LBXSKSI), serum uric acid (LBDSUASI), serum aspartate aminotransferase (LBXSATSI), serum alanine aminotransferase (LBXSASSI), serum creatinine (LBDSCRSI), and serum albumin (LBDSALSI). All biochemical indicators were measured through standardized laboratory tests according to NHANES laboratory protocols. At the same time, we included diabetes and hypertension as covariates in the analysis. For diabetes diagnosis, we combined self-reported data from participants (DIQ010) with the use of insulin (DIQ050) and antidiabetic medications (DIQ070), and included glycated hemoglobin (LBXGH) to improve the detection rate of diabetes. Diabetes was diagnosed when glycated hemoglobin ≥ 6.5%. For hypertension diagnosis, we combined self-reported data (BPQ020) with the use of antihypertensive medications (BPQ040A, BPQ050A), and calculated the average of three blood pressure measurements (BPXSY1-BPXSY3, BPXDI1-BPXDI3) to improve the diagnosis rate of hypertension. Hypertension was diagnosed when systolic blood pressure ≥ 130 or diastolic blood pressure ≥ 90. The selection of these covariates was based on their potential impact on metabolic health, cardiovascular disease risk, and other chronic conditions [22,23].

## 2.5. Statistical analysis

Given the complex sampling design of the NHANES survey, survey weights were applied to ensure national representativeness of the results [24]. Continuous variables were expressed as weighted means with standard deviations, while categorical variables were presented as frequencies and proportions [25,26]. Baseline characteristics across different HOMA-IR, TyG and TyG-BMI index quartile groups were compared using t-tests for continuous variables and chi-square tests for categorical variables. HOMA-IR, TyG and TyG-BMI index scores were divided into quartiles, with the first quartile (Q1) serving as the reference group.

Multivariable logistic regression was used to analyze the association between HOMA-IR, TyG, TyG-BMI and the prevalence of HF, including both unadjusted models and progressively adjusted models. Model I adjusted for sex, age, and race/ethnicity. Model II further adjusted for hypertension, total cholesterol, low-density lipoprotein cholesterol, and serum sodium levels. Model III additionally adjusted for diabetes. Model IV further incorporated adjustments for serum potassium, serum uric acid, alanine aminotransferase (ALT), aspartate aminotransferase (AST), creatinine, and albumin. Model V included all covariates [27,28].

To examine the nonlinear relationship between HOMA-IR, TyG, TyG-BMI and HF prevalence, restricted cubic spline (RCS) regression analysis was performed, with knots placed at the 10th, 50th, and 90th percentiles [29]. Stratified analyses were conducted based on age, sex, race/ethnicity, diabetes status, and hypertension. Interaction P-values were calculated using likelihood ratio tests in multivariable logistic regression (LRT) models to assess the consistency of relationships across subgroups. A predictive nomogram model incorporating key HF-related variables was developed using multivariable logistic regression, and its discriminative ability was evaluated using receiver operating characteristic (ROC) curves with the area under the curve (AUC) calculation and decision curve analysis (DCA) to assess clinical utility.

To ensure the accurate representativeness of the NHANES 2011−2016 data, we have adjusted the 2-year weight (WTMEC2YR) by dividing it by 3, as per the NHANES guidelines (WTMEC6YR).Missing covariates were handled using the "multiple imputation" method to avoid bias from excluding participants with incomplete data [30,31]. All statistical analyses were performed using R software (version 4.1.6, R Foundation for Statistical Computing, Vienna, Austria), and a P-value < 0.05 was considered statistically significant. The main R packages used in this study included: dplyr (version 1.1.4) for data manipulation and cleaning, ggplot2(version 3.5.1) for data visualization, mice (version 3.16.0) for multiple imputation of missing data, rms (version 6.8−2) for constructing nomogram models, pROC (version 1.18.5) for ROC curve analysis, and car (version 3.1−3) for multivariable regression analysis. These R packages and versions ensure the transparency and reproducibility of the analysis.

## 3. Results

### 3.1. Characteristics of the study population

In this study, a total of 29,902 participants from NHANES (2011–2016) were included. Based on the established inclusion and exclusion criteria, 7,668 adults with complete information (insulin, fasting glucose, triglycerides, and HF diagnosis) were ultimately included, representing a diverse U.S. population of over 100 million individuals. The average age of the study population was 48.25 years. A total of 3,727 participants were male (48.36% of the weighted population). Non-Hispanic Whites represented the largest racial group, comprising 2,964 individuals (66.50% of the weighted population). A total of 284 participants had heart failure, corresponding to a weighted prevalence of 3.01%.

Participants with a history of HF were generally older (66.16 years vs. 47.70 years, P < 0.001) and had higher prevalences of diabetes (36.29% vs. 11.93%, P < 0.001) and hypertension (87.32% vs. 48.08%, P < 0.001). Participants in the HF group exhibited higher levels of potassium (P < 0.001), uric acid (P < 0.001), creatinine (P < 0.001), and albumin (P < 0.001) compared with those without HF. Interestingly, participants in the HF group had relatively lower lipid levels, including total cholesterol (P < 0.001), LDL-C (P < 0.001), and HDL-C (P < 0.001). Detailed demographic and clinical characteristics are presented in Table 1.

HOMA-IR, TyG and TyG-BMI were calculated based on insulin, fasting glucose, and triglycerides. The findings indicate that participants in the HF group had higher levels of fasting glucose (P < 0.001), fasting triglycerides (P < 0.01), and fasting insulin (P < 0.001), corresponding to higher HOMA-IR (P < 0.001),TyG and TyG-BMI (P < 0.001).

### 3.2. HOMA-IR, TyG and TyG-BMI associated with increased risk of HF

This study explored the association between HOMA-IR, TyG and TyG-BMI with the risk of HF through logistic regression analysis. We categorized HOMA-IR (Q1: < 1.47; Q2: 1.47–2.44; Q3: 2.44–4.35; Q4: ≥ 4.35), TyG (Q1: < 8.11; Q2: 8.11–8.54; Q3: 8.54–8.99; Q4: ≥ 8.99) and TyG-BMI (Q1: < 201; Q2: 201–239; Q3: 239–286; Q4: ≥ 286) into four quartiles (Table 2). The results indicated that HOMA-IR, TyG and TyG-BMI were significantly associated with an increased risk of HF.

In the unadjusted model, HOMA-IR as a continuous variable demonstrated that for each unit increase, the risk of HF increased by 1.7% (OR = 1.017, 95% CI = 1.006–1.027, P < 0.01). Additionally, quartile analysis of HOMA-IR revealed that compared to the lowest quartile (Q1), the highest quartile (Q4) was significantly associated with an increased risk of HF in the unadjusted model (OR = 2.577, 95% CI = 1.639–4.050, P < 0.001). However, as the models were adjusted, the

**Table 1. Baseline characteristics of the study population stratified by HF status.**

| Variables | Overall (n = 7668) | Non- HF (n = 7384) | HF (n = 284) | P value |
|---|---|---|---|---|
| Age, years | 48.25[47.49,49.01] | 47.70[46.95,48.44] | 66.16[64.15,68.18] | <0.001*** |
| ≥60 years | 28.43[26.66,30.20] | 27.03[25.27,28.78] | 73.56[66.09,81.04] | |
| <60 years | 71.57[69.80,73.34] | 72.97[71.22,74.73] | 26.44[19.10,33.91] | |
| Sex-male, % | | | | 0.94 |
| Male, % | 48.36[47.11,49.60] | 48.35[47.00,49.69] | 48.70[39.93,57.42] | |
| Female, % | 51.65[50.40,52.89] | 51.66[50.31,53.00] | 51.33[42.58,60.07] | |
| Race, % | | | | 0.12 |
| Mexican American, % | 8.20[6.17,10.23] | 8.30[6.25,10.35] | 4.94[1.94,7.94] | |
| Other Hispanic, % | 6.25[4.66,7.85] | 6.30[4.72,7.88] | 4.94[2.48,7.40] | |
| Non-Hispanic White, % | 66.50[62.36,70.64] | 66.33[62.17,70.48] | 72.05[66.71,77.39] | |
| Non-Hispanic Black, % | 10.61[8.42,12.80] | 10.57[8.40,12.75] | 12.10[8.09,16.11] | |
| Other Race, % | 8.43[7.19,9.67] | 8.51[7.29,9.73] | 5.96[1.15,10.77] | |
| DM, % | 12.66[11.47,13.84] | 11.93[10.76,13.09] | 36.29[29.50,43.09] | <0.001*** |
| Hypertension, % | 49.26[47.20,51.32] | 48.08[45.99,50.17] | 87.32[82.83,91.82] | <0.001*** |
| TC, mmol/L | 4.98[4.94,5.01] | 4.99[4.95,5.03] | 4.56[4.39,4.73] | <0.001*** |
| LDL-C, mmol/L | 2.90[2.87,2.93] | 2.91[2.88,2.94] | 2.51[2.37,2.65] | <0.001*** |
| HDL-C, mmol/L | 1.42[1.40,1.44] | 1.43[1.41,1.45] | 1.27[1.20,1.33] | <0.001*** |
| Na, mmol/L | 139.11[138.97,139.25] | 139.11[138.97,139.25] | 139.08[138.71,139.44] | 0.87 |
| K, mmol/L | 4.01[3.99,4.03] | 4.01[3.99,4.02] | 4.14[4.10,4.18] | <0.001*** |
| UA, µmol/L | 324.04[321.54,326.54] | 322.36[319.66,325.06] | 378.25[360.23,396.27] | <0.001*** |
| AST, U/L | 25.34[24.92,25.76] | 25.32[24.90,25.74] | 26.13[21.54,30.72] | 0.73 |
| ALT, U/L | 25.62[25.21,26.03] | 25.51[25.11,25.90] | 29.27[24.39,34.15] | 0.14 |
| Cr, µmol/L | 77.67[76.78,78.56] | 76.95[76.04,77.86] | 101.05[94.59,107.51] | <0.001*** |
| Alb, g/L | 42.88[42.74,43.02] | 42.93[42.79,43.08] | 41.14[40.60,41.68] | <0.001*** |
| Triglycerides, mg/dL | 125.99[122.28,129.70] | 125.16[121.31,129.01] | 152.68[135.65,169.71] | <0.01** |
| Insulin, µU/mL | 13.35[12.83,13.86] | 13.21[12.69,13.74] | 17.66[15.47,19.85] | <0.001*** |
| Glucose, mg/dL | 106.92[105.92,107.92] | 106.43[4.62,4.65] | 122.78[116.88,128.68] | <0.001*** |
| HOMA-IR | 3.89[3.70,4.08] | 3.82[3.63,4.02] | 5.97[4.87,7.07] | <0.001*** |
| TyG | 8.58[8.55,8.61] | 8.57[8.54,8.60] | 8.89[8.77,9.01] | <0.001*** |
| TyG-BMI | 251.04[247.70,254.38] | 250.03[246.76,253.30] | 283.83[269.15,298.55] | <0.001*** |

Continuous variables are presented as means with 95% confidence intervals, while categorical variables are presented as proportions with 95% confidence intervals.

TC: Total cholesterol; LDL-C: Low-density lipoprotein cholesterol; HDL-C: High-density lipoprotein cholesterol; Na: Serum sodium; K: Serum potassium; UA: Uric acid; AST: Aspartate aminotransferase; ALT: Alanine aminotransferase; Cr: Creatinine; Alb: Albumin.

*** p < 0.001, ** p < 0.01, * p < 0.05.

association between HOMA-IR and HF risk gradually weakened. Notably, in model 5, the third quartile (Q3) even showed a protective effect against HF risk (OR = 0.560, 95% CI = 0.356–0.882, P < 0.01), suggesting the possibility of a nonlinear relationship.

Analysis of TyG showed that it was significantly associated with the risk of HF in both the unadjusted model and models 1–3. In the unadjusted model, each unit increase in TyG was associated with a 1.8-fold increase in the risk of HF (OR = 1.798, 95% CI: 1.453–2.225, P < 0.001). In the quartile analysis, compared to the first quartile (Q1), the second, third, and fourth quartiles (Q2, Q3, Q4) all showed varying degrees of increased risk. Notably, the highest

**Table 2. Weighted Logistic Regression Analysis of the Associations between HOMA-IR, TyG and TyG-BMI Index with HF Risk.**

| | Non-adjusted Model OR [95% CI] | P value | Model 1 OR [95% CI] | P value | Model 2 OR [95% CI] | P value |
|---|---|---|---|---|---|---|
| HOMA.IR | 1.017[1.006 - 1.027] | <0.01 | 1.012[1.005 - 1.020] | <0.01 | 1.010[1.002 - 1.018] | <0.05 |
| HOMA.IR-Q1 | Reference | – | Reference | – | Reference | – |
| HOMA.IR-Q2 | 0.965[0.612 - 1.520] | 0.878 | 0.810[0.523 - 1.255] | 0.351 | 0.780[0.491 - 1.240] | 0.301 |
| HOMA.IR-Q3 | 1.166[0.739 - 1.838] | 0.513 | 0.931[0.596 - 1.454] | 0.755 | 0.832[0.523 - 1.323] | 0.442 |
| HOMA.IR-Q4 | 2.577[1.639 - 4.050] | <0.001 | 2.064[1.326 - 3.213] | <0.01 | 1.563[0.988 - 2.472] | 0.065 |
| TyG | 1.798[1.453 - 2.225] | <0.001 | 1.670[1.306 - 2.136] | <0.001 | 1.849[1.374 - 2.488] | <0.001 |
| TyG-Q1 | Reference | – | Reference | – | Reference | – |
| TyG-Q2 | 1.563[0.913 - 2.674] | 0.110 | 1.197[0.681 - 2.104] | 0.536 | 1.198[0.691 - 2.076] | 0.525 |
| TyG-Q3 | 2.199[1.426 - 3.393] | <0.001 | 1.606[1.022 - 2.524] | <0.05 | 1.615[1.047 - 2.492] | 0.037 |
| TyG-Q4 | 3.571[2.279 - 5.596] | <0.001 | 2.478[1.585 - 3.875] | <0.001 | 2.590[1.627 - 4.125] | <0.001 |
| TyG-BMI | 1.006[1.004 - 1.008] | <0.001 | 1.007[1.004 - 1.009] | <0.001 | 1.006[1.003 - 1.008] | <0.001 |
| TyG-BMI-Q1 | Reference | – | Reference | – | Reference | – |
| TyG-BMI-Q2 | 0.988[0.627 - 1.559] | 0.961 | 0.673[0.422 - 1.074] | 0.105 | 0.625[0.392 - 0.996] | 0.056 |
| TyG-BMI-Q3 | 1.519[0.856 - 2.696] | 0.160 | 1.198[0.703 - 2.041] | 0.511 | 1.077[0.630 - 1.839] | 0.788 |
| TyG-BMI-Q4 | 2.667[1.619 - 4.395] | <0.01 | 2.337[1.379 - 3.959] | <0.01 | 1.885[1.092 - 3.256] | 0.029 |
| | Model 3 OR [95% CI] | P value | Model 4 OR [95% CI] | P value | Model 5 OR [95% CI] | P value |
| HOMA.IR | 1.006[0.998 - 1.014] | 0.163 | 1.002[0.992 - 1.012] | 0.766 | 0.997[0.986 - 1.008] | 0.562 |
| HOMA.IR-Q1 | Reference | – | Reference | – | Reference | – |
| HOMA.IR-Q2 | 0.774[0.487 - 1.231] | 0.287 | 0.691[0.440 - 1.084] | 0.119 | 0.634[0.395 - 1.017] | 0.070 |
| HOMA.IR-Q3 | 0.799[0.506 - 1.260] | 0.341 | 0.670[0.437 - 1.025] | 0.076 | 0.560[0.356 - 0.882] | <0.05 |
| HOMA.IR-Q4 | 1.396[0.863 - 2.255] | 0.183 | 1.007[0.611 - 1.659] | 0.680 | 0.745[0.425 - 1.304] | 0.312 |
| TyG | 1.765[1.260 - 2.473] | <0.01 | 1.505[1.109 - 2.041] | 0.979 | 0.925[0.606 - 1.412] | 0.720 |
| TyG-Q1 | Reference | – | Reference | – | Reference | – |
| TyG-Q2 | 1.176[0.668 - 2.068] | 0.588 | 1.116[0.625 - 1.992] | 0.713 | 0.971[0.552 - 1.709] | 0.919 |
| TyG-Q3 | 1.553[0.987 - 2.443] | 0.066 | 1.377[0.845 - 2.243] | 0.210 | 1.039[0.644 - 1.678] | 0.876 |
| TyG-Q4 | 2.354[1.365 - 4.060] | <0.01 | 1.931[1.097 - 3.399] | <0.05 | 1.081[0.555 - 2.107] | 0.820 |
| TyG-BMI | 1.005[1.002 - 1.008] | <0.01 | 1.003[1.000 - 1.006] | 0.070 | 1.002[0.998 - 1.005] | 0.406 |
| TyG-BMI-Q1 | Reference | – | Reference | – | Reference | – |
| TyG-BMI-Q2 | 0.614[0.381 - 0.990] | 0.054 | 0.548[0.335 - 0.898] | <0.05 | 0.477[0.293 - 0.774] | <0.01 |
| TyG-BMI-Q3 | 1.033[0.606 - 1.764] | 0.905 | 0.851[0.513 - 1.410] | 0.536 | 0.642[0.368 - 1.122] | 0.132 |
| TyG-BMI-Q4 | 1.727[0.956 - 3.121] | 0.080 | 1.152[0.617 - 2.150] | 0.661 | 0.781[0.389 - 1.570] | 0.494 |

Data are presented as odds ratios (OR) with 95% confidence intervals (CI). Model 1: Adjusted for gender, age, and race/ethnicity. Model 2: Further adjusted for hypertension, total cholesterol, low-density lipoprotein cholesterol (LDL-C), and serum sodium. Model 3: Additionally adjusted for diabetes. Model 4: Further included potassium, uric acid, alanine aminotransferase (ALT), aspartate aminotransferase (AST), creatinine, and albumin. Model 5: Adjusted for all covariates listed above.

quartile (Q4) was significantly associated with an elevated risk of HF (OR = 3.571, 95% CI: 2.279–5.596, P < 0.001). Even after model adjustments, TyG remained a strong predictor. In contrast, TyG-BMI exhibited similar predictive stability to TyG, showing a significant association with HF risk in both the unadjusted model and models 1–3. However, the association between TyG-BMI and HF was not as strong. For example, in the unadjusted model, each unit increase in TyG-BMI was associated with only a 0.6% increase in HF risk (OR = 1.006, 95% CI: 1.004–1.008, P < 0.001).

### 3.3. Association of HOMA-IR, TyG and TyG-BMI with HF risk in different subgroups

This study evaluated the relationship between HOMA-IR and the TyG index with HF risk across various population characteristics, using data from the NHANES 2011–2016. The analysis revealed that factors such as sex, race, age, diabetes, and hypertension were significantly associated with HOMA-IR, TyG, and TyG-BMI index.

For HOMA-IR, an increase was significantly associated with HF risk in both males (OR: 1.729, 95% CI: 1.001–1.015, P<0.05) and females (OR: 1.905, 95% CI: 1.021–1.055, P<0.001), with the interaction suggesting that it may be more sensitive in females (P<0.05). Race analysis revealed that non-Hispanic whites had an OR of 2.600 (95% CI: 1.440–4.695, P<0.01) in the Q4 group, while non-Hispanic blacks exhibited a significant risk association, with an OR reaching 3.558 (95% CI: 1.912–6.620, P<0.001). Other racial groups, including multiracial individuals, had an OR of 2.110 (95% CI: 0.341–13.072) in the Q4 group, which, although relatively high, had weaker statistical significance.

In exploring the association between the TyG index and HF risk, we found significant heterogeneity across different population characteristics. For females in the Q4 group, the odds ratio was 3.674 (95% CI: 1.986–6.795, P<0.001), while for males it was 3.499 (95% CI: 1.728–7.082, P<0.001), indicating that both genders face elevated HF risk with increased TyG index levels. Racial analysis showed that non-Hispanic whites had an OR of 3.728 (95% CI: 1.983–7.010, P<0.001) in the Q4 group, whereas non-Hispanic blacks exhibited an even higher OR of 4.101 (95% CI: 2.321–7.248, P<0.001), suggesting heightened susceptibility to HF risk in these populations. Age analysis revealed that the OR for individuals aged ≥60 years in the Q4 group was 2.297 (95% CI: 1.381–3.823), while those aged <60 years reached an OR of 4.116 (95% CI: 1.910–8.874), indicating significant risk among younger individuals. Additionally, the study showed that the TyG index in non-diabetic individuals had an OR of 1.585 (95% CI: 1.168–2.151, P<0.01), and in the Q4 group, the OR reached 2.863 (95% CI: 1.634–5.018, P<0.001). The interaction suggests that TyG may be more sensitive in non-diabetic patients (P<0.05). For hypertension, hypertensive patients in the Q4 group had an OR of 2.750 (95% CI: 1.768–4.277, P<0.0001), while the association was weaker for individuals without hypertension (P>0.05). The overall predictive ability of TyG-BMI was similar to that of TyG; however, in diabetic individuals where TyG showed poorer predictive ability, TyG-BMI demonstrated potential, with an OR of 1.003 (95% CI: 1.000–1.005, P<0.05).

Detailed results for continuous variables, quartile analysis, and subgroup analyses regarding both HOMA-IR, TyG and TyG-BMI are presented in Table 3.

### 3.4. Comparison of the effectiveness of HOMA-IR, TyG and TyG-BMI in predicting HF risk

In this study, we systematically assessed the effectiveness of HOMA-IR, TyG, and TyG-BMI in predicting HF risk. Through ROC curve analysis, we found that the area under the curve (AUC) for TyG-BMI (AUC=0.62) was higher than that for TyG (AUC=0.61), which was higher than HOMA-IR (AUC=0.60), indicating the superiority of TyG-BMI and TyG in identifying HF risk (Fig 2). Additionally, decision curve analysis (DCA) further confirmed the clinical applicability of TyG and TyG-BMI across different decision thresholds, demonstrating their higher decision value (Fig 3). In the restricted cubic spline (RCS) curve analysis, both TyG (Fig 4) and TyG-BMI (Fig 5) exhibited a positive correlation trend with HF risk, while HOMA-IR did not demonstrate the same prognostic capability (Fig 6).

## 4. Discussion

This study explored the effectiveness of HOMA-IR, TyG, and TyG-BMI in predicting HF risk, revealing significant differences in their associations with HF across various population characteristics. Our analysis highlights the potential advantages of TyG in HF risk assessment, while also emphasizing the relevance of HOMA-IR and TyG-BMI in specific subgroups.

HF is a common condition in cardiovascular diseases, characterized by impaired cardiac pumping function, leading to inadequate tissue and organ perfusion. The occurrence of HF is closely related to various risk factors, including

**Table 3. Overall and subgroup analyses for the associations between HOMA-IR, TyG, TyG-BMI index (Continuous Variables and Quartile Analysis).**

| | HOMA-IR | P value | TyG | P value | TyG-BMI | P value |
|---|---|---|---|---|---|---|
| Gender | OR [95% CI] | Pint<0.01 | OR [95% CI] | Pint=0.662 | OR [95% CI] | Pint=0.187 |
| Female | 1.038 [1.021 - 1.055] | <0.0001 | 1.905 [1.336 - 2.718] | <0.001 | 1.005 [1.002 - 1.007] | <0.001 |
| Q2 | 0.958 [0.445 - 2.062] | 0.912 | 1.794 [0.830 - 3.879] | 0.144 | 1.017 [0.613 - 1.689] | 0.948 |
| Q3 | 0.930 [0.413 - 2.092] | 0.861 | 2.238 [1.227 - 4.082] | 0.011804 | 1.775 [0.769 - 4.100] | 0.186 |
| Q4 | 2.386 [1.192 - 4.777] | <0.05 | 3.674 [1.986 - 6.795] | <0.001 | 2.035 [1.032 - 4.014] | <0.05 |
| Male | 1.008 [1.001 - 1.015] | <0.05 | 1.729 [1.360 - 2.199] | <0.0001 | 1.007 [1.004 - 1.010] | <0.0001 |
| Q2 | 0.977 [0.510 - 1.873] | 0.944 | 1.307 [0.640 - 2.668] | 0.466 | 0.992 [0.492 - 2.000] | 0.982 |
| Q3 | 1.470 [0.869 - 2.487] | 0.158 | 2.164 [1.102 - 4.253] | <0.05 | 1.335 [0.751 - 2.373] | 0.33 |
| Q4 | 2.851 [1.582 - 5.140] | <0.01 | 3.499 [1.728 - 7.082] | <0.01 | 3.595 [1.931 - 6.693] | <0.001 |
| Race | | Pint=0.437 | | Pint=0.238 | | Pint=0.182 |
| Mexican American | 1.008 [0.992 - 1.024] | 0.355 | 1.254 [0.762 - 2.065] | 0.379 | 1.008 [1.004 - 1.011] | <0.001 |
| Q2 | 0.540 [0.071 - 4.089] | 0.555 | 2.094 [0.238 - 18.423] | 0.51 | 0.565 [0.109 - 2.921] | 0.5 |
| Q3 | 0.903 [0.198 - 4.111] | 0.896 | 1.678 [0.303 - 9.289] | 0.557 | 1.235 [0.280 - 5.448] | 0.783 |
| Q4 | 1.831 [0.664 - 5.048] | 0.251 | 2.923 [0.332 - 25.714] | 0.341 | 1.956 [0.634 - 6.035] | 0.252 |
| Other Hispanic | 1.011 [1.000 - 1.023] | 0.052 | 2.039 [0.989 - 4.206] | 0.061 | 1.006 [1.001 - 1.012] | <0.05 |
| Q2 | 0.563 [0.147 - 2.153] | 0.407 | 1.245 [0.314 - 4.946] | 0.757 | 2.349 [0.468 - 11.788] | 0.306 |
| Q3 | 0.445 [0.142 - 1.398] | 0.174 | 0.892 [0.206 - 3.857] | 0.879 | 1.160 [0.272 - 4.954] | 0.842 |
| Q4 | 2.117 [0.852 - 5.261] | 0.115 | 2.377 [0.728 - 7.764] | 0.16 | 4.228 [1.149 - 15.562] | <0.05 |
| Non-Hispanic White | 1.023 [1.008 - 1.037] | <0.01 | 1.766 [1.331 - 2.342] | <0.001 | 1.005 [1.002 - 1.008] | <0.001 |
| Q2 | 0.948 [0.538 - 1.669] | 0.854 | 1.720 [0.770 - 3.843] | 0.193 | 0.810 [0.449 - 1.460] | 0.487 |
| Q3 | 1.157 [0.634 - 2.112] | 0.636 | 2.605 [1.390 - 4.883] | <0.01 | 1.473 [0.706 - 3.075] | 0.308 |
| Q4 | 2.600 [1.440 - 4.695] | <0.01 | 3.728 [1.983 - 7.010] | <0.001 | 2.352 [1.246 - 4.439] | <0.05 |
| Non-Hispanic Black | 1.015 [1.004 - 1.027] | <0.05 | 1.869 [1.405 - 2.486] | <0.001 | 1.006 [1.003 - 1.009] | <0.001 |
| Q2 | 1.347 [0.550 - 3.298] | 0.519 | 1.489 [0.844 - 2.627] | 0.177 | 1.624 [0.695 - 3.792] | 0.27 |
| Q3 | 0.866 [0.289 - 2.593] | 0.799 | 1.796 [0.875 - 3.687] | 0.119 | 1.515 [0.704 - 3.256] | 0.294 |
| Q4 | 3.558 [1.912 - 6.620] | <0.001 | 4.101 [2.321 - 7.248] | <0.0001 | 2.482 [1.049 - 5.871] | <0.05 |
| Other Race | 1.014 [0.992 - 1.037] | 0.229 | 3.246 [1.974 - 5.335] | <0.0001 | 1.012 [1.003 - 1.021] | <0.01 |
| Q2 | 1.856 [0.402 - 8.578] | 0.433 | 0.516 [0.082 - 3.234] | 0.484 | 1.793 [0.389 - 8.260] | 0.458 |
| Q3 | 4.614 [0.937 - 22.721] | 0.067 | 2.183 [0.366 - 13.022] | 0.396 | 3.501 [0.737 - 16.630] | 0.122 |
| Q4 | 2.110 [0.341 - 13.072] | 0.427 | 7.205 [1.602 - 32.411] | <0.05 | 12.922 [2.593 - 64.388] | <0.01 |
| Age | | Pint=0.420 | | Pint=0.653 | | Pint=0.912 |
| ≥60 | 1.011 [1.002 - 1.021] | <0.05 | 1.667 [1.255 - 2.214] | <0.001 | 1.006 [1.004 - 1.008] | <0.0001 |
| Q2 | 1.199 [0.732 - 1.965] | 0.475 | 1.017 [0.598 - 1.729] | 0.951 | 0.789 [0.429 - 1.453] | 0.451 |
| Q3 | 1.393 [0.919 - 2.111] | 0.125 | 1.632 [0.917 - 2.905] | 0.103 | 1.335 [0.732 - 2.437] | 0.351 |
| Q4 | 2.195 [1.455 - 3.312] | <0.001 | 2.297 [1.381 - 3.823] | <0.01 | 2.342 [1.387 - 3.955] | <0.01 |
| <60 | 1.017 [1.006 - 1.029] | <0.01 | 1.827 [1.282 - 2.604] | <0.01 | 1.006 [1.003 - 1.010] | <0.01 |
| Q2 | 0.386 [0.133 - 1.118] | 0.086 | 2.380 [0.797 - 7.106] | 0.127 | 0.728 [0.403 - 1.315] | 0.299 |
| Q3 | 0.404 [0.145 - 1.120] | 0.088 | 2.147 [0.807 - 5.717] | 0.133 | 1.352 [0.536 - 3.409] | 0.527 |
| Q4 | 2.366 [1.121 - 4.997] | <0.05 | 4.116 [1.910 - 8.874] | <0.001 | 2.364 [0.941 - 5.939] | 0.074 |
| Diabetes | | Pint=0.070 | | Pint<0.05 | | Pint=0.334 |
| Yes | 0.998 [0.987 - 1.010] | 0.783 | 1.097 [0.851 - 1.414] | 0.479 | 1.003 [1.000 - 1.005] | <0.05 |
| Q2 | 1.798 [0.596 - 5.420] | 0.303 | 1.355 [0.551 - 3.333] | 0.511 | 1.456 [0.418 - 5.077] | 0.558 |
| Q3 | 1.366 [0.524 - 3.563] | 0.527 | 1.243 [0.518 - 2.982] | 0.628 | 2.987 [0.971 - 9.190] | 0.063 |
| Q4 | 1.522 [0.655 - 3.536] | 0.335 | 1.184 [0.582 - 2.410] | 0.643 | 2.799 [0.963 - 8.135] | 0.065 |
| No | 1.038 [0.999 - 1.079] | 0.062 | 1.585 [1.168 - 2.151] | <0.01 | 1.004 [1.001 - 1.008] | <0.05 |

*(Continued)*

**Table 3.** (Continued)

| | HOMA-IR | *P* value | TyG | *P* value | TyG-BMI | *P* value |
|---|---|---|---|---|---|---|
| Q2 | 0.798 [0.475 - 1.341] | 0.399 | 1.407 [0.734 - 2.696] | 0.309 | 0.862 [0.515 - 1.441] | 0.573 |
| Q3 | 0.867 [0.517 - 1.453] | 0.591 | 1.910 [1.119 - 3.262] | <0.05 | 1.059 [0.576 - 1.947] | 0.854 |
| Q4 | 1.878 [1.091 - 3.234] | <0.05 | 2.863 [1.634 - 5.018] | <0.001 | 1.811 [0.965 - 3.398] | 0.071 |
| Hypertension | | Pint = 0.695 | | Pint = 0.273 | | Pint = 0.942 |
| Yes | 1.012 [1.001 - 1.022] | <0.05 | 1.518 [1.178 - 1.957] | <0.01 | 1.004 [1.001 - 1.006] | <0.01 |
| Q2 | 0.946 [0.552 - 1.619] | 0.839 | 1.678 [0.917 - 3.074] | 0.1 | 0.754 [0.484 - 1.174] | 0.218 |
| Q3 | 0.981 [0.563 - 1.707] | 0.946 | 2.018 [1.261 - 3.228] | <0.01 | 1.183 [0.630 - 2.221] | 0.604 |
| Q4 | 1.596 [0.865 - 2.946] | 0.142 | 2.750 [1.768 - 4.277] | <0.0001 | 1.490 [0.895 - 2.479] | 0.132 |
| No | 1.015 [1.002 - 1.029] | <0.05 | 0.897 [0.381 - 2.113] | 0.805 | 1.003 [0.996 - 1.010] | 0.355 |
| Q2 | 0.435 [0.119 - 1.590] | 0.215 | 0.565 [0.159 - 2.004] | 0.381 | 0.863 [0.282 - 2.640] | 0.798 |
| Q3 | 0.270 [0.099 - 0.739] | <0.05 | 0.848 [0.234 - 3.076] | 0.804 | 0.332 [0.098 - 1.132] | 0.085 |
| Q4 | 2.234 [0.756 - 6.599] | 0.153 | 1.105 [0.271 - 4.503] | 0.89 | 2.184 [0.681 - 7.009] | 0.196 |

Data are presented as odds ratios (OR) with 95% confidence intervals (CI). Pint represents the P-value for the interaction effect between the outcome. The interaction term was tested for statistical significance, with P < 0.05 considered significant.

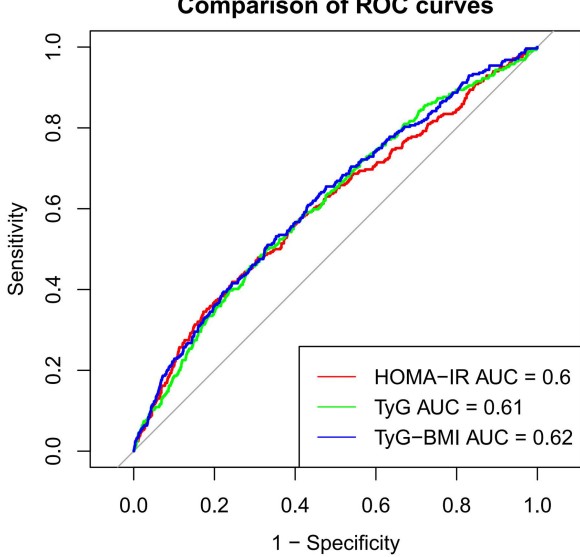

**Fig 2. ROC curve analysis of HOMA-IR, TyG and TyG-BMI index.**

hypertension, coronary artery disease, cardiomyopathy, diabetes, and metabolic syndrome [32–34]. IR is considered one of the key mechanisms in HF, potentially affecting cardiac function through multiple pathways [6,8]. HOMA-IR, TyG, and TyG-BMI are key indicators for assessing IR and play a crucial role in predicting HF risk. Research suggests that the TyG index is closely associated with HF incidence, reflecting lipid metabolism abnormalities and IR status, thus revealing potential mechanisms underlying HF risk. HOMA-IR effectively evaluates the degree of IR, with its increasing values significantly associated with HF risk [9,14]. TyG-BMI, as a novel composite index, further enhances the predictive ability of HF risk assessment. TyG-BMI combines the influences of lipid metabolism, IR, and body mass index, which may provide a more comprehensive reflection of cardiovascular health [35].

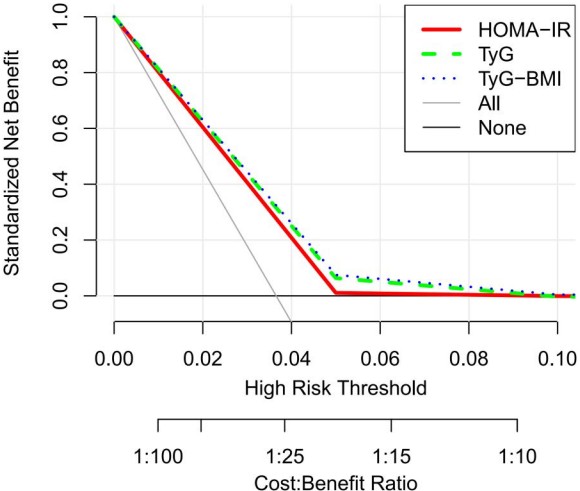

**Fig 3. DCA curve analysis of HOMA-IR, TyG and TyG-BMI index.**

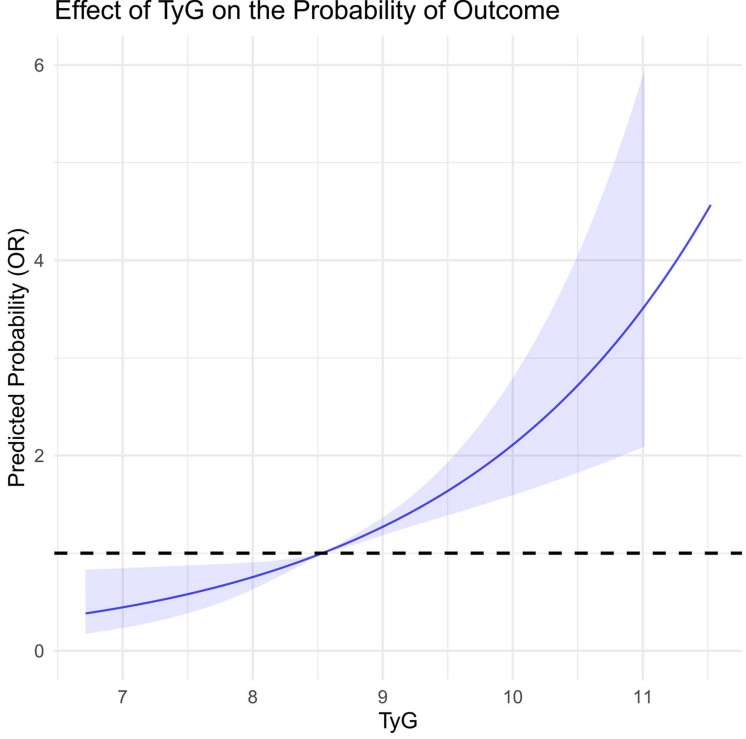

**Fig 4. RCS curve analysis of the TyG index.**

Our study found that HOMA-IR, TyG, and TyG-BMI are all significantly associated with HF risk, with TyG showing the strongest association. In the unadjusted model, the risk of HF in the Q4 group for TyG was nearly 4 times higher and showed stability in multiple models. However, the predictive ability of HOMA-IR decreased with model adjustments. TyG-BMI's predictive stability was similar to that of TyG, but its OR approached 1, indicating weaker association with HF

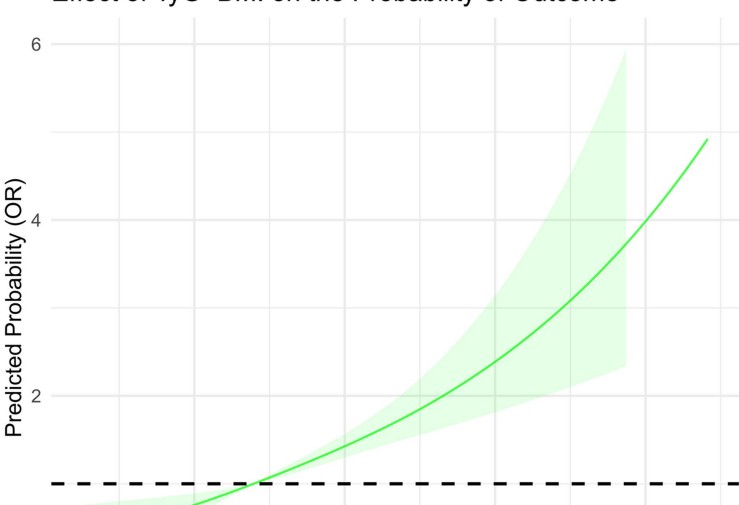

**Fig 5. RCS curve analysis of the TyG-BMI index.**

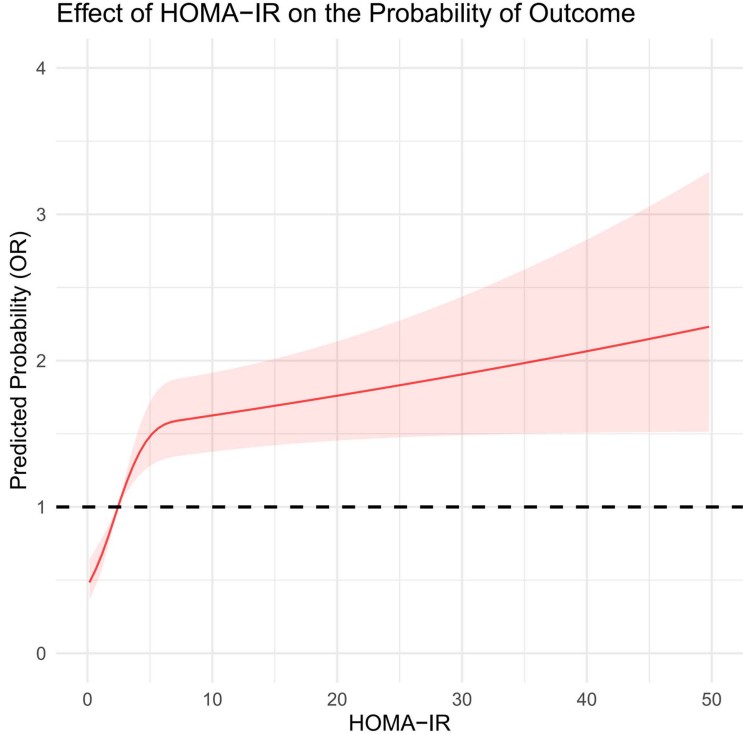

**Fig 6. RCS curve analysis of the HOMA-IR index.**

risk. In subgroup analyses, we found that the presence of diabetes seemed to affect the predictive performance of TyG. Previous studies suggest that metabolic factors such as diabetes and insulin treatment may influence the TyG index [36]. However, TyG-BMI appears to compensate for this effect. The inclusion of body mass index may reduce the potential impact of diabetes treatment on metabolic indices. A prospective cohort study in an Asian population examining the long-term relationship between TyG and HF risk showed that after adjusting for potential confounders, Cox regression analysis revealed hazard ratios of 1.02, 1.29, and 1.40 for Q2, Q3, and Q4 groups, respectively. Subgroup analyses also showed a significant interaction between cumulative TyG index and BMI or waist circumference [10]. These findings are consistent with our results.

Recent research shows that IR can lead to cardiac metabolic disturbances, potentially causing myocardial cell dysfunction [37]. This process is likely associated with inflammatory responses, lipotoxicity, and oxidative stress. Under normal physiological conditions, insulin helps maintain energy balance in myocardial cells and promotes their growth and survival through signaling pathways such as PI3K and MAPK [38]. However, in an IR state, these pathways are suppressed, leading to impaired myocardial function and ultimately to HF [39]. Additionally, HF patients often present with metabolic syndrome, further exacerbating the cardiac burden. Recent studies increasingly emphasize the importance of early identification and intervention of HF risk factors, especially in high-risk populations with IR and metabolic syndrome [40]. Therefore, combining multiple biomarkers, such as TyG, HOMA-IR, and TyG-BMI, with clinical indicators for assessing HF risk may offer new directions for improving patient prognosis.

We further assessed the predictive sensitivity and specificity of HOMA-IR, TyG, and TyG-BMI using ROC curves, DCA curves, and RCS analysis. The AUCs showed that TyG and TyG-BMI have advantages in risk identification. DCA curve analysis also supports the clinical applicability of the TyG index at different decision thresholds, highlighting its potential as a risk prediction tool. TyG-BMI also demonstrated similar advantages. In subgroup analyses by gender and age, the associations between TyG index and TyG-BMI with HF risk showed different predictive abilities across populations. This suggests that clinical risk assessment strategies need to be adjusted based on different population characteristics. Notably, we found that the relationship between HOMA-IR and HF risk may not be linear. Some studies suggest that increasing IR could lead to cardiac metabolic disturbances and impaired cardiac function, but this process may not increase proportionally [41]. The effect could intensify as the severity of IR increases. Mild IR may mainly affect metabolic health with minimal direct impact on the heart, whereas severe IR may significantly exacerbate cardiac load through inflammatory responses, fatty acid metabolism imbalance, and oxidative stress, thus increasing the risk of heart failure. Therefore, the relationship between IR and HF risk may not be simple linear, but rather exhibit varying effects depending on the degree of IR [42].

Mechanistically, the relationship between HOMA-IR and TyG index with HF risk may involve multiple biological mechanisms. IR may exacerbate the occurrence of HF by promoting inflammatory responses, increasing the release of free fatty acids from adipocytes, and causing myocardial metabolic imbalance [43]. Additionally, the close relationship between the TyG index and metabolic syndrome may provide new insights into HF development. This mechanism may be related to endothelial dysfunction and the progression of atherosclerosis, thus affecting cardiac blood supply and function. Recent epidemiological studies have clearly revealed a significant association between IR and the risk of HF occurrence [44]. This mechanism may involve chronic inflammatory responses, endothelial dysfunction, and disturbances in fatty acid metabolism, which in turn impact myocardial energy metabolism and function. Moreover, the presence of metabolic syndrome further exacerbates cardiovascular risk, potentially influencing HF development through mechanisms such as accelerated atherosclerosis and increased cardiac load [45].

Despite the important insights provided by this study, there are some limitations. These include: (1) Population applicability: The study used data from the NHANES database. Although this dataset offers broad population representation, its specific characteristics of the U.S. population may affect the generalizability of the results. Specifically, since NHANES data is mainly from the U.S., it may not fully represent populations from other countries or regions. (2) Self-reported bias: NHANES data relies on self-reports, especially for disease diagnoses and lifestyle-related data. This could introduce

information bias, as self-reported data may not be entirely accurate or may be subject to recall bias. (3) Potential impact of diabetes on IR indices: Diabetes, especially in patients receiving insulin treatment, may affect the calculation of HOMA-IR and TyG indices. In this study, insulin treatment in diabetic patients may alter their metabolic characteristics, influencing the accuracy of IR indices. Although we considered diabetes as a factor in the analysis, the diversity of the diabetic population, including differences in diabetes type, control status, and medication use, was not fully controlled, which could lead to different metabolic characteristics in this group compared to non-diabetic patients. (4)Lack of HF treatment information: This study did not collect treatment information for HF patients, including medication use, surgical interventions, and other clinical management measures. HF treatment significantly impacts patient prognosis, so the lack of consideration of treatment interventions may affect the assessment of HF risk. Future studies should incorporate clinical data to further validate the findings of this study. (5) Limitations of cross-sectional design: As a cross-sectional study, this research can only reveal the association between HOMA-IR, TyG, and TyG-BMI with HF, but cannot establish causality. Therefore, although we observed a strong correlation between these metabolic indices and HF risk, it does not imply that they are direct causes of HF occurrence.

## 5. Conclusions

This study evaluated the effectiveness of HOMA-IR, TyG, and TyG-BMI in predicting HF risk. The results indicate that TyG has a stronger association with HF risk than HOMA-IR and TyG-BMI, and it outperforms HOMA-IR in predicting HF risk, especially in specific subgroups. Our findings highlight the potential of the TyG index as a tool for HF risk assessment, warranting further exploration of its mechanisms and clinical applications to improve early identification and management of HF.

## Supporting information

**S1 File. The complete characteristics of the study population.**
(CSV)

**S2 File. Adjustments and subgroup analyses for HOMA-IR regression analysis model.**
(CSV)

**S3 File. Adjustments and subgroup analyses for TyG regression analysis model.**
(CSV)

**S4 File. Adjustments and subgroup analyses for TyG-BMI regression analysis model.**
(CSV)

## Acknowledgments

We would like to express our gratitude to the National Health and Nutrition Examination Survey (NHANES) team for their efforts in data collection and management. We also thank our colleagues for their support and constructive feedback throughout the study. Additionally, we appreciate the funding support that made this research possible. Finally, we would like to acknowledge the participants of the NHANES study, as their contributions are invaluable to our research.

## Author contributions

**Conceptualization:** Meijuan Lu, Junchi Guo.

**Data curation:** Meijuan Lu, Junchi Guo.

**Formal analysis:** Meijuan Lu, Junchi Guo.

**Funding acquisition:** Mingyan Zhang.

**Investigation:** Junchi Guo.

**Methodology:** Junchi Guo.

**Project administration:** Junchi Guo.

**Resources:** Meijuan Lu, Mingyan Zhang.

**Software:** Meijuan Lu, Peili Yang, Teng Ma, Mingyan Zhang.

**Supervision:** Junchi Guo.

**Validation:** Junchi Guo.

**Visualization:** Meijuan Lu, Junchi Guo.

**Writing – original draft:** Meijuan Lu, Junchi Guo, Mingyan Zhang.

**Writing – review & editing:** Meijuan Lu, Junchi Guo, Peili Yang, Teng Ma, Mingyan Zhang.

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
