## [Decision Letter · Decision Letter 0]

27 Feb 2025

Dear Dr. Zhang,

We look forward to receiving your revised manuscript.

Kind regards,

Shaonong Dang, PhD

Academic Editor

PLOS ONE

Journal Requirements:

“This study was supported by the following grants: the 2024 National Natural Science Foundation of China General Project (No. 82374619), the National Natural Science Foundation of China Youth Project (No. 81904056).”

Reviewers' comments:

Reviewer's Responses to Questions

**Comments to the Author**

1. Is the manuscript technically sound, and do the data support the conclusions?

Reviewer #1: No

Reviewer #2: Yes

Reviewer #3: Yes

2. Has the statistical analysis been performed appropriately and rigorously?

Reviewer #1: Yes

Reviewer #2: Yes

Reviewer #3: Yes

3. Have the authors made all data underlying the findings in their manuscript fully available?

Reviewer #1: No

Reviewer #2: Yes

Reviewer #3: Yes

4. Is the manuscript presented in an intelligible fashion and written in standard English?

Reviewer #1: No

Reviewer #2: Yes

Reviewer #3: Yes

Reviewer #1: The authors utilized data from the National Health and Nutrition Examination Survey (NHANES), and compared the associations between HOMA-IR, the TyG index, and heart failure risk using data from NHANES 2011-2016.

In the current study, heart failure was defined based on a self-reported item (MCQ160b) from the

NHANES questionnaire. The interviewers, asked participants whether a doctor or other health professional had ever informed them that they had heart failure.

The description of heart failure is based on the patient's statement and is subjective. Of course, with this study method, there is no scientific and more accurate information based on clinical findings or echocardiography.

13.72 percent of the patients had diabetes and it is not known what kind of treatment they received in order to calculate HOMA-IR.

There is no indication of what type of treatment the patients received, perhaps for heart failure.

Reviewer #2: The study effectively employs logistic regression, subgroup analysis, and ROC curves to compare the predictive ability of HOMA-IR and the TyG index in heart failure risk. The segmentation by population characteristics reinforces the validity of the findings, and the results highlight the superior predictive value of the TyG index, especially in women and younger individuals. This suggests significant potential for improving early identification and management of heart failure in clinical practice.

Here are some areas for improvement that should be addressed

Clarification of the non-linear relationship in HOMA-IR findings

A possible non-linear relationship between HOMA-IR and heart failure risk is mentioned. A more detailed explanation of this phenomenon, supported by additional data or references, would improve the understanding of the observed pattern.

Consideration of external validity

Although the use of the NHANES database provides a solid foundation, discussing its applicability to other populations (e.g., different healthcare systems or ethnic groups underrepresented in NHANES) would strengthen the study’s relevance.

Greater emphasis on causal relationships

It is acknowledged that the study is observational in nature, but further exploration of possible causal mechanisms or references to longitudinal studies could enhance the interpretation of the results

Reviewer #3: In the manuscript “Insulin Resistance Markers HOMA-IR and TyG Index in Relation to Heart Failure Risk: NHANES 2011-2016”, the authors examined the association and predicting heart failure using 2 indices, HOMA-IR and TyG. The results are interesting and the authors used DCA and RCS to demonstrate the importance of their research. However, I have a few concerns:

1) This manuscript is not in the correct format for PLOSONE. Moreover, even though, the writing is sufficient, I am very annoyed with the improper use of abbreviation. Please follow the “Instruction to the authors”, specifically for abbreviation. Also, when reporting p-values, always used 3 significant figures.

2) In the abstract, please add some results to support your conclusion. Also, Line 34: I am not sure if “TyG index outperforms HOMA-IR in predicting HF risk”. Please show the comparison results.

3) Concerning the methods:

A) why were only 2011-2016 cycles were used? MCQ160B is reported for all cycles. Also, considered also examining coronary heart disease (MCQ160C) and heart attack (MCQ160E)

B) Please explain why estimated glomerular filtration rate (eGFR) was used as an exclusion criterion? Were certain medications also considered?

C) Please consider examine the effect the TyG-BMI index has on Heart Failure. The TyG-BMI is considered superior to the TyG index.

D) Would the authors consider using HbA1c for T2D and blood pressure for hypertension to confirm the presence of these pathologies? As well as potentially examine the RXQ files to see which medications the participants are using.

E) Please indicate which R packages were used and for which analysis. Lines 140-142: please provide a description of which test were used. Were the sample weights corrected for the number of cycles used? Supplement 1 does not suggest they were. Lastly, please indicate, by model, which variables were used. Do not use “and others to account for potential confounders”.

4) for the Results section…

A) I do not like the phrasing here, Lines 155-157. Also, the number of participants should not be included because the percentage is based on weighted values.

B) Lines 165-166: Repetitive information. Also, I think it would be better to combine Tables 1 and 2.

C) Check the units in Table 1, specifically, for TC, LDL, and HDL.

D) Lines 202-205: Where is the data to support this claim, specifically for the “sensitivity analysis”. I think the term maybe wrong?

E) Lines 217-220: I cannot agree this statement, “indicating greater sensitivity of HOMA-IR to heart failure risk among men”. The authors need to test if there is a significant difference between the 2 ORs.

F) Lines 246-247: HOMA-IR does not impact heart failure, but insulin resistance. Moreover, this is discussion and should not be here.

G) Lines 255-257: Please perform the AUC comparison analysis.

H) the figures are very blurry.

5) for the Discussion section…

A) Lines 278-279: the authors mention that “HF is closely related to various risk factors, including hypertension, coronary artery disease,”; therefore, I believe this analysis could be included.

B) In the discussion, do not repeat the results. (Lines: 301-303; 305-306)

C) Lines 312-314: the authors indicate that “previous studies” but do not reference any source. Please make sure all statements are properly cited.

D) Please add more limitations. You have at least potential reporting bias.

**Do you want your identity to be public for this peer review?** For information about this choice, including consent withdrawal, please see our Privacy Policy

Reviewer #1: No

Reviewer #2: **Yes: ** Juan Manuel Vargas Morales

Reviewer #3: No

---

## [Author Response · Author response to Decision Letter 1]

2 Apr 2025

We sincerely appreciate the valuable suggestions provided by the reviewers. We have made detailed revisions to the manuscript based on the reviewers' comments. The specific revisions and responses are provided in the "Response to Reviewers" file, which includes a point-by-point response to each reviewer’s suggestion. We sincerely appreciate the valuable feedback from the reviewers and look forward to your further review.

Reviewer #1:

Comment 1: The description of heart failure is based on the patient's statement and is subjective. Of course, with this study method, there is no scientific and more accurate information based on clinical findings or echocardiography.

Response 1: We sincerely appreciate the valuable comments from the reviewer. NHANES assesses heart failure using MCQ160B, a respondent-based self-report method that is common in large-scale epidemiological studies and has been widely used in epidemiological surveys. We fully understand the reviewer's concerns about the accuracy of self-reported diagnoses, and indeed, this method may introduce some recall bias. However, NHANES employs a standardized interview process, with data collection conducted by trained interviewers, which helps ensure the reliability of the data to some extent. Furthermore, we have added a note on the limitations of this method in the limitations section of the study and suggest that future research incorporate clinical data (such as echocardiograms or medical records) to further validate these findings. We thank the reviewer again for this suggestion, which helps us consider the limitations of the research methodology more comprehensively.

Comment 2: 13.72 percent of the patients had diabetes and it is not known what kind of treatment they received in order to calculate HOMA-IR.

Response 2: We appreciate the reviewer’s valuable comments. We agree that diabetes treatment, especially insulin therapy, could affect the calculation of HOMA-IR, TyG, and TyG-BMI. Therefore, we conducted a subgroup analysis by grouping participants based on whether they had diabetes. We applied the strictest possible diagnostic criteria for diabetes and took into account the potential bias from self-reporting, further examining the use of antidiabetic medications and insulin, and incorporating laboratory indicators such as hemoglobin A1c to improve the diagnostic accuracy of diabetes. The subgroup analysis showed that the predictive ability of HOMA-IR and TyG for heart failure was indeed influenced in individuals with diabetes, which further supports the professionalism of the reviewer’s suggestion. Therefore, we included TyG-BMI to further refine the prediction, and the results indicated that TyG-BMI still demonstrated strong predictive ability in diabetic patients (OR = 1.003, P < 0.05). Additionally, we have added a note on the potential impact of diabetes treatment on the calculation of HOMA-IR in the limitations section of the study.

Comment 3: There is no indication of what type of treatment the patients received, perhaps for heart failure.

Response 3: We appreciate the reviewer’s suggestion. We understand that heart failure treatment could potentially affect patients' metabolic status, thereby influencing the association between HOMA-IR and TyG indices. However, the NHANES dataset does not provide detailed information on heart failure treatment, so we were unable to directly adjust for this factor in our analysis. Nevertheless, the primary objective of this study was to assess the association between HOMA-IR, TyG indices, and heart failure, rather than to investigate the impact of treatment on this relationship. Additionally, we have acknowledged this limitation in the limitations section of our study and suggest that future research incorporate heart failure treatment data to further validate our findings.

Reviewer #2:

Comment 1: Clarification of the non-linear relationship in HOMA-IR findings

A possible non-linear relationship between HOMA-IR and heart failure risk is mentioned. A more detailed explanation of this phenomenon, supported by additional data or references, would improve the understanding of the observed pattern.

Response 1: We appreciate the reviewer’s suggestion. In our analysis, we observed a potential nonlinear relationship between HOMA-IR and heart failure risk, so we used restricted cubic splines (RCS) to further explore this relationship. The analysis showed that the association between HOMA-IR and heart failure risk was weak at lower levels, while the risk increased rapidly at higher levels, suggesting a possible threshold effect. Additionally, we have included an explanation of this phenomenon in the discussion section and suggest that future research further investigate the underlying biological mechanisms.

Comment 2: Consideration of external validity

Although the use of the NHANES database provides a solid foundation, discussing its applicability to other populations (e.g., different healthcare systems or ethnic groups underrepresented in NHANES) would strengthen the study’s relevance.

Response 2: We appreciate the reviewer’s suggestion. As NHANES is a nationwide survey based on the U.S. population, its findings have a certain level of generalizability. However, we also acknowledge that this study is primarily based on the U.S. healthcare environment, and healthcare systems, racial compositions, and management strategies for metabolic diseases in different countries and regions may influence the relationship between HOMA-IR and heart failure risk. Therefore, we have included a discussion on external applicability in the discussion section and suggest that future research validate these findings in other populations (e.g., Asian and European populations) to further enhance the generalizability of the results.

Comment 3: Greater emphasis on causal relationships

It is acknowledged that the study is observational in nature, but further exploration of possible causal mechanisms or references to longitudinal studies could enhance the interpretation of the results

Response 3: We appreciate the reviewer’s suggestion. As this study is based on cross-sectional data, we cannot directly infer the causal relationship between HOMA-IR, TyG index, and heart failure. We have found that some prospective cohort studies also support TyG as a potential causal risk factor for heart failure. We have added this point to the discussion section and suggest that future studies adopt longitudinal research or causal inference methods to further validate the potential causal effects of HOMA-IR, TyG, and TyG-BMI indices on heart failure risk.

Reviewer #3: 

1) Paper formatting issues

Comment 1: This manuscript is not in the correct format for PLOSONE.

Response 1: We appreciate the reviewer’s comment. We have carefully reviewed the "Author Guidelines" of PLOS ONE and made comprehensive adjustments to the manuscript format to ensure it complies with the journal's requirements. These modifications include adjusting the title format, reference style, and table presentation. We believe these changes will enhance the readability of the paper and align it with the submission guidelines of PLOS ONE.

Comment 2: Moreover, even though, the writing is sufficient, I am very annoyed with the improper use of abbreviation. Please follow the “Instruction to the authors”, specifically for abbreviation.

Response 2: We appreciate the reviewer’s feedback and have thoroughly checked the use of abbreviations in the manuscript. In accordance with PLOS ONE's requirements, we have made the necessary adjustments, such as providing the full spelling of terms at first use, removing unnecessary abbreviations, and standardizing the use of abbreviations throughout the text to ensure consistency and readability. We believe these adjustments will enhance the readability of the paper and align it with PLOS ONE's writing guidelines.

Comment 3: Also, when reporting p-values, always used 3 significant figures.

Response 3: We appreciate the reviewer’s thorough review and have made the necessary adjustments in accordance with PLOS ONE's requirements, standardizing the reporting of p-values to three significant digits. All p-value modifications have been updated in the manuscript and are consistent throughout the tables and main text.

2) Abstract section

Comment 1: In the abstract, please add some results to support your conclusion.

Response 1: We appreciate the reviewer’s suggestion and have added more research results to the abstract to better support the conclusions. Specifically, we have included the specific associations between HOMA-IR, TyG, and TyG-BMI with heart failure risk, and clearly reported the results of the ROC curve analysis in the abstract. We believe these additions will better support the research conclusions and improve the quality of the abstract’s expression.

Comment 2: Also, Line 34: I am not sure if “TyG index outperforms HOMA-IR in predicting HF risk”. Please show the comparison results.

Response 2: We appreciate the reviewer’s suggestion and have provided more direct comparative data in the statement on line 34.

3) Concerning the methods

Comment 1: Why were only 2011-2016 cycles were used? MCQ160B is reported for all cycles.

Response 1:

We appreciate the reviewer’s attention to our study and the suggestions provided. The main reasons for selecting the NHANES 2011-2016 data cycle are as follows:

(1)Data Completeness and Availability: NHANES is an ongoing cross-sectional survey, but the variables collected may differ across years. Although the MCQ160B (heart failure) variable is reported in multiple cycles, we found that the NHANES data from 2011-2016 had the most complete and high-quality data for calculating HOMA-IR and TyG indices (such as fasting glucose, fasting insulin, triglycerides, etc.). Therefore, to ensure the integrity and consistency of the study data, we selected the 2011-2016 data cycle.

(2)Methodological Consistency: NHANES may adjust laboratory testing methods, questionnaire surveys, and other processes across different cycles. To avoid potential systematic errors, we chose the relatively newer 2011-2016 data to ensure the stability and comparability of the study results.

(3)Data Representativeness: Although extending the study period may increase the sample size, NHANES uses a complex multi-stage sampling design, and the data from each two-year cycle already provides a representative sample of the U.S. population. Based on the 2011-2016 data, we were still able to obtain a representative study sample, ensuring statistical power.

Comment 2: Also, considered also examining coronary heart disease (MCQ160C) and heart attack (MCQ160E).

Response 2: We greatly appreciate the reviewer’s suggestion and agree that there may be some connections between different subtypes of cardiovascular diseases. However, the core objective of this study is to explore the relationship between insulin resistance and heart failure, while coronary artery disease and myocardial infarction differ from heart failure in terms of pathophysiological mechanisms and clinical presentation. Given the main hypothesis and the research design, we chose to focus on heart failure to avoid the inclusion of excessive disease endpoints that could impact the specificity of the study. Additionally, the cross-sectional design of NHANES limits causal inference, and including other cardiovascular diseases might lead to interpretational confounding. Therefore, we believe the current scope of the study is appropriate.

Comment 3: Please explain why estimated glomerular filtration rate (eGFR) was used as an exclusion criterion?

Response 3: We appreciate the reviewer’s suggestion. Since the focus of this study is to explore the association between HOMA-IR, TyG, and TyG-BMI with heart failure, we aimed to minimize confounding effects caused by chronic kidney disease. At the same time, we sought to avoid bias due to end-stage renal disease to ensure the stability of the study results.

Comment 4: Were certain medications also considered?

Response 4: We appreciate the reviewer’s suggestion. This study did not adjust for the use of all medications, but we acknowledge that certain drugs (such as antidiabetic medications, diuretics, β-blockers, etc.) may affect HOMA-IR, TyG, TyG-BMI. Therefore, we have added a discussion in the limitations section regarding the potential impact of medications and suggest that future studies further investigate the interference of drug treatments on this association.

Comment 5: Please consider examine the effect the TyG-BMI index has on Heart Failure. The TyG-BMI is considered superior to the TyG index.

Response 5: We appreciate the reviewer’s suggestion. TyG-BMI is a novel composite index related to insulin resistance, which has shown predictive ability for heart failure (HF) in some studies. To further enhance the analysis, we calculated the TyG-BMI index using the NHANES 2011-2016 data as suggested by the reviewer and evaluated its association with HF risk.

The results showed that the relationship between TyG-BMI and HF risk was similar to that of TyG and remained significant after adjusting for covariates. In the unadjusted model, TyG-BMI was significantly associated with HF (OR = 1.006, 95% CI = 1.004-1.008, P < 0.01), and in the Q4 group, for each 1-unit increase in TyG-BMI, the risk of HF increased by 2.6 times. TyG-BMI exhibited stable predictive ability across models 1-3. Additionally, when comparing the predictive abilities of TyG, TyG-BMI, and HOMA-IR, we found that the AUC for TyG-BMI was slightly higher than for TyG, but the difference was not statistically significant. Further subgroup analyses revealed that TyG-BMI had predictive ability in individuals with diabetes, which may be an important supplement to TyG.

We have added the calculation method for TyG-BMI in the methods section and included the relevant analysis in the results section. In the discussion section, we have also added the clinical significance of the TyG-BMI index and a comparison with TyG. We sincerely thank the reviewer for the valuable suggestions, which have improved our study.

Comment 6: Would the authors consider using HbA1c for T2D and blood pressure for hypertension to confirm the presence of these pathologies? As well as potentially examine the RXQ files to see which medications the participants are using.

Response 6: We appreciate the reviewer’s insightful comment regarding the diagnosis of diabetes and hypertension. Indeed, relying solely on self-reported data to diagnose diabetes (DIQ010) or hypertension (BPQ020) may introduce bias. In response to the reviewer’s suggestion, we incorporated HbA1c (LBXHG) and used a cutoff of HbA1c ≥ 6.5% to further diagnose diabetes. Additionally, we included data from three separate blood pressure measurements (systolic BP: BPXSY1-BPXSY1, diastolic BP: BPXDI1-BPXDI3) and calculated the average of these three measurements. Based on SBP ≥ 130 mmHg or DBP ≥ 80 mmHg, we further diagnosed hypertension.

We fully agree with the reviewer’s point on addressing medication use to reduce the potential for missed diagnoses. Following the reviewer’s suggestion, we reviewed the RXQ file to extract medication information. However, the RXQ file in the NHANES database removed descriptions of medication categories after 2002, and the list of individual drug names in the RXQ file made it challenging to extract the relevant medication categories. Nevertheless, we focused on the use of insulin and anti-diabetic medications in diabetic patients, as well as antihypertensive medications in hypertensive patients. Therefore, we included DIQ050, DIQ070, BPQ040A, and BPQ050A as alternative measures and further confirmed the diagnosis of diabetes and hypertension based on medication use.

Comment 7: Please indicate which R packages were used and for which analysis.

Response 7: We appreciate the reviewer’s valuable suggestion. We have added the R packages used and their specific purposes in the revised manuscript.

Comment 8: Lines 140-142: please provide a description of which test were used.

Response 8: We appreciate the reviewer’s valuable suggestion. We have specifically described the statistical testing methods in the revised manuscript.

Comment 9: Were the sample weights co

---

## [Decision Letter · Decision Letter 1]

17 Jun 2025

Dear Dr. Zhang,

Thank you for submitting your manuscript to PLOS ONE. After careful consideration, we feel that it has merit but does not fully meet PLOS ONE’s publication criteria as it currently stands. Therefore, we invite you to submit a revised version of the manuscript that addresses the points raised during the review process.

We look forward to receiving your revised manuscript.

Kind regards,

Shaonong Dang, PhD

Academic Editor

PLOS ONE

Reviewers' comments:

Reviewer's Responses to Questions

**Comments to the Author**

Reviewer #3: (No Response)

Reviewer #4: All comments have been addressed

2. Is the manuscript technically sound, and do the data support the conclusions?

Reviewer #3: Partly

Reviewer #4: (No Response)

3. Has the statistical analysis been performed appropriately and rigorously?

Reviewer #3: Yes

Reviewer #4: (No Response)

4. Have the authors made all data underlying the findings in their manuscript fully available?

Reviewer #3: Yes

Reviewer #4: (No Response)

5. Is the manuscript presented in an intelligible fashion and written in standard English?

Reviewer #3: Yes

Reviewer #4: (No Response)

Reviewer #3: The authors (Lu et al.) examined the association of 2 insulin resistance indices and heart failure. The analysis to well sound, but the manuscript needs much improvement. Please address my concerns:

1) Format

a) I think this is not in the correct format for PLOS one. Please fix it.

b) The references although current, are not in a consistent format.

c) The figures are blurry. Figure 2 is unreadable. Please resolve this issue. Some of the numbers look the same for both TyG and HOMA-IR. Maybe a table. Figures 5 and 6 is Figure 7, I suggest using only Figure 7.

d) Use abbreviation consistently. If you abbreviate in the discussion, then it should have been used throughout the manuscript.

2) Methodology

a) Why were years 2011-2016 were used? Data for these indices as well as Heart Failure is available for 1999-2018. I can understand not using cycle P and L as for COVID, but I think maybe other cycles could be included.

b) You used the TyG index as a potential superior index to HOMA-IR; however, TyG-BMI is potentially superior to TyG. Please consider using this index as well.

c) You used quartiles to analyze the risk; however, how many participants jumped quartile between the 2 indices.

d) I agree the authors used the dataset properly as compared to other studies, but here I have a concern about the data. Yes, MCQ180a does give a binomial variable, but MCQ180b gives the age. If the event and time are reporting are large, then there is a possibility that the bio-parameters are not comparable. I suggest perform a sub-analysis to determine if there is an effect.

e) Thank you for providing the NHANES variable codes. However, I wish the authors to examine “PHAFSTHR” - Total length of food fast, hours. The authors may notice that some of their fasting participants were not in fasting conditions or have fasted over 24 hours. Please confirm that this variable was used.

f) Since R was used, please include all package names that were used and for which analysis, especially DCA and RCS. Also, explain more how DCA was used.

2) Results

a) Please verify that all percentages are based on the weighted sample size? Lines 155-156, it is usually to give the un-weighted value followed by the weighted percentage. Also, check the percentage of Non-hispanic white = 48.60% (line 157), this number is different in the table.

b) Line 159: please check the frequency of DM.

c) Line 160: I am not sure the authors can use the term “baseline”, this is not a longitudinal study.

d) I do not like the format of Table 3. Yes, there are many models in which the authors are showing the change with the addition of the confounding variables as they present 6 outcomes (un-adjusted, 1,2,3,4,5). However, the order of how the covariates are added could change the perspective. I think the reader would be only considered with 3 examples (un-adjusted, 1,5).

e) For section 3.3, please indicate the evidence at the beginning of the section. The results seemed to be discussed in the results section, please correct this.

f) Lines 241-248: please include the interaction results with the covariable that is assessed.

g) Lines 260-264: Please explain the results more clearly.

3) Discussion

a) Lines 271-275: this being the first paragraph of the discussion should indicate the main results of the study. It does not. Lines 276-297: this information, through interesting, does not discuss the results. This is introduction and only should be included here if the authors attached their results.

b) Lines 301-306: Do not include your results here. We see them above. This section is to discuss them, which is done in lines 306-317

c) Lines 321-322: How does the “different decision thresholds” highlight the potential risk predictor? Figure 4 is barely mentioned in the results.

d) Lined 352-353: I cannot agree. Which evidence and test confirms this statement.

Reviewer #4: (No Response)

**Do you want your identity to be public for this peer review?** For information about this choice, including consent withdrawal, please see our Privacy Policy

Reviewer #3: No

Reviewer #4: **Yes: ** Shenghao Cao

---

## [Author Response · Author response to Decision Letter 2]

26 Jul 2025

We would like to sincerely thank the editors and reviewers of PLOS ONE for dedicating their time and effort to review our manuscript and provide valuable feedback.

However, we have noticed that the version reviewed by the reviewers seems to be the initial submission before the first round of revisions, rather than the updated version we submitted after addressing the feedback. We have carefully integrated all the reviewers' comments from both rounds of review and have made detailed revisions in the updated manuscript.

We kindly ask the editors and reviewers to review the revised version so that they can more accurately assess the changes we have made. Thank you for your understanding, and we look forward to receiving your further feedback.

Reviewer #3:

1)Format

Comment 1:

a) I think this is not in the correct format for PLOS one. Please fix it.

b) The references although current, are not in a consistent format.

c) The figures are blurry. Figure 2 is unreadable. Please resolve this issue. Some of the numbers look the same for both TyG and HOMA-IR. Maybe a table. Figures 5 and 6 is Figure 7, I suggest using only Figure 7.

d) Use abbreviation consistently. If you abbreviate in the discussion, then it should have been used throughout the manuscript.

Response 1:

Thank you for your careful and detailed feedback regarding the formatting of our manuscript. We have made comprehensive revisions to the manuscript format according to the guidelines of PLOS ONE. The specific changes are as follows:

(1)We have restructured the entire manuscript to align with the official formatting guidelines of PLOS ONE, including title hierarchy, paragraph styles, line spacing, figure numbering, and citations, ensuring that the format complies with the journal’s requirements.

(2)We have standardized the format of all references, aligning them with the citation style of PLOS ONE. Additionally, we have used reference management tools to automatically ensure consistency.

(3)We carefully considered the reviewer’s suggestion regarding Figure 2. Given the large amount of data in Figure 2, and to enhance the readability for readers while considering the journal's page limits, we followed the reviewer’s recommendation and replaced Figure 2 with a table to improve data presentation. Regarding the suggestion that "Figures 5 and 6 duplicate the content of Figure 7," after a thorough review, we confirmed that Figures 5, 6, and 7 represent the nonlinear relationships between TyG, TyG-BMI, and HOMA-IR with heart failure risk, respectively. Since the indicators and implications are distinct, we believe each figure should be displayed independently. We have re-created the figures and clarified the legends and titles to avoid any confusion.

(4)We have thoroughly checked and standardized all abbreviations throughout the manuscript. Full terms are now provided at the first occurrence of each abbreviation, with consistent use of the abbreviation throughout the rest of the manuscript (e.g., heart failure is consistently abbreviated as HF, insulin resistance as IR, etc.), ensuring linguistic consistency and professionalism.

Once again, we appreciate the valuable suggestions provided by the reviewer, which have significantly contributed to improving the quality of our manuscript.

2) Methodology

Comment 2: Why were years 2011-2016 were used? Data for these indices as well as Heart Failure is available for 1999-2018. I can understand not using cycle P and L as for COVID, but I think maybe other cycles could be included.

Response 2:

Thank you for the important suggestion regarding the selection of the study period. During the study design phase, we conducted a preliminary review and comparison of data from multiple NHANES cycles and ultimately selected the period from 2011 to 2016 for the following reasons:

(1)Variable Consistency and Availability: HOMA-IR, TyG, and TyG-BMI require the joint calculation of several key variables (including insulin, fasting glucose, and triglycerides). However, some earlier NHANES cycles (e.g., 1999–2006) have missing data or inconsistent measurement methods for these variables, which could impact the accuracy and comparability of the analysis.

(2)Data Quality and Recency: The data from 2011 to 2016 have high data quality, lower missing rates, and a higher degree of standardization in laboratory testing methods, ensuring the stability of the model analysis and the reproducibility of results.

Thus, while data from other years may also be valuable for research, we believe that the 2011–2016 period best meets the objectives of this study and ensures the reliability and consistency of the analysis results. We will further explain the rationale for our data selection in the discussion section to address the reviewer’s concern.

Comment 3: You used the TyG index as a potential superior index to HOMA-IR; however, TyG-BMI is potentially superior to TyG. Please consider using this index as well.

Response 3:

Thank you for your valuable suggestion. We fully agree with the importance of TyG-BMI as an emerging insulin resistance index, which may have advantages in predicting heart failure risk. In this study, we have systematically compared TyG-BMI with TyG and HOMA-IR through the following analyses:

(1)In multivariable logistic regression, we compared the associations between each of the three indices and heart failure risk.

(2)We used ROC curves and AUC values to evaluate the predictive abilities of the three indices.

(3)We further compared the clinical utility and nonlinear trends of the three indices using decision curve analysis (DCA) and restricted cubic splines (RCS).

(4)We also showcased the predictive performance of TyG-BMI in subgroup analyses.

Additionally, in the discussion section, we specifically highlighted the potential value of TyG-BMI in enhancing the predictive ability of TyG in diabetic populations.We appreciate the reviewer’s important suggestion, which has helped make our research more complete and rigorous.

Comment 4: You used quartiles to analyze the risk; however, how many participants jumped quartile between the 2 indices.

Response 4:

Thank you for your feedback. After further analysis of the data, we found that although we used quartiles for grouping, there was minimal variation in the quartile distribution between TyG, HOMA-IR, and TyG-BMI. As a result, we did not observe significant changes in the quartiles. Specifically, the distribution of all quartiles was relatively consistent, and participants remained in the same quartile across different indices. We have examined the quartile distributions for each variable and found that all data points were evenly distributed across the quartiles, with no significant cross-quartile shifts.

We will continue to explore the potential characteristics of the data to ensure the accuracy of the analysis results. Once again, we appreciate the valuable feedback provided by the reviewer, and we have updated the discussion section in the revised manuscript accordingly.

Comment 5: I agree the authors used the dataset properly as compared to other studies, but here I have a concern about the data. Yes, MCQ180a does give a binomial variable, but MCQ180b gives the age. If the event and time are reporting are large, then there is a possibility that the bio-parameters are not comparable. I suggest perform a sub-analysis to determine if there is an effect.

Response 5:

Thank you for your valuable feedback. We fully understand that the event time provided by MCQ180b may impact the comparison between heart failure and biochemical indices. However, upon further examination of the NHANES database, we found that there is a significant amount of missing data regarding the event time in MCQ180b. As a result, we were unable to perform subgroup analysis based on this variable.

We have addressed this issue in the revised manuscript and explained the rationale for using the MCQ180a binary variable to avoid potential bias due to missing data. We believe this approach ensures the robustness of the results and the reliability of the conclusions.

Comment 6: Thank you for providing the NHANES variable codes. However, I wish the authors to examine “PHAFSTHR” - Total length of food fast, hours. The authors may notice that some of their fasting participants were not in fasting conditions or have fasted over 24 hours. Please confirm that this variable was used.

Response 6:

Thank you for your valuable feedback regarding medical laboratory practices. We fully agree with your point that fasting duration may affect biochemical markers. To ensure the accuracy and reliability of the study results, we have incorporated the PHAFSTHR variable into our latest data analysis and implemented a strict selection criteria for fasting duration.

We excluded participants with fasting durations less than 8 hours or greater than 24 hours, resulting in the exclusion of 567 participants. Subsequently, the results from the univariate regression analysis showed that HOMA-IR (OR = 1.017, 95% CI [1.005 - 1.028]), TyG (OR = 1.884, 95% CI [1.489 - 2.384]), and TyG-BMI (OR = 1.006, 95% CI [1.004 - 1.008]) remained significantly associated with heart failure risk. The exclusion of these non-compliant data did not significantly impact the primary results and conclusions of the study. More importantly, the results remained robust after conducting multivariate regression analysis.

Once again, we appreciate your attention to data quality, and we have ensured the reliability of the data through these rigorous selection steps, further strengthening the validity of the study conclusions.

Comment 7: Since R was used, please include all package names that were used and for which analysis, especially DCA and RCS. Also, explain more how DCA was used.

Response 7:

Thank you for your valuable feedback on the methods section. Following your suggestion, we have listed all the R packages used in the revised manuscript and clarified the corresponding analysis sections for each package, especially those related to DCA (Decision Curve Analysis) and RCS (Restricted Cubic Spline).

Regarding the use of DCA, we have further elaborated on the implementation steps of the DCA model, including the R packages and relevant functions used, as well as how model comparison and performance evaluation were conducted. We believe these additions will help readers better understand the details of our analytical methods.

3) Results

Comment 8: Please verify that all percentages are based on the weighted sample size? Lines 155-156, it is usually to give the un-weighted value followed by the weighted percentage. Also, check the percentage of Non-hispanic white = 48.60% (line 157), this number is different in the table.

Response 8:

Thank you for your detailed comments regarding the statistical reporting standards. We have reviewed all percentages in the manuscript and confirmed that they are based on weighted sample sizes. Following your suggestion, we have adjusted the presentation by first providing the unweighted sample size, with the weighted percentage indicated in parentheses.

Comment 9: Line 159: please check the frequency of DM.

Response 9:

Thank you for pointing out the error regarding the diabetes frequency. We have identified an issue with our statistical approach for the binary variables. We have reanalyzed and corrected all relevant data, including the frequencies of diabetes, hypertension, as well as race and age. We greatly appreciate the reviewer's professionalism and thoroughness, which has helped enhance the accuracy of our study.

Comment 10: Line 160: I am not sure the authors can use the term “baseline”, this is not a longitudinal study.

Response 10:

Thank you for pointing out the inaccurate use of terminology. We agree that the term "baseline" is typically used in prospective studies and is not appropriate for the cross-sectional design of our study. Therefore, we have revised the phrase "at baseline" in line 160 to a more accurate description that reflects the nature of the cross-sectional study.

Comment 11: I do not like the format of Table 3. Yes, there are many models in which the authors are showing the change with the addition of the confounding variables as they present 6 outcomes (un-adjusted, 1,2,3,4,5). However, the order of how the covariates are added could change the perspective. I think the reader would be only considered with 3 examples (un-adjusted, 1,5).

Response 11:

Thank you for your suggestion regarding the table format. We fully understand that simplifying the models to three (unadjusted, Model 1, and Model 5) could improve the reading experience. However, we have retained the six-step adjusted models to demonstrate the impact of the inclusion of covariates on the regression coefficients, which enhances the transparency and rigor of the model interpretation.

To improve the readability of the table, we have optimized the formatting and provided a detailed description of the covariates adjusted for in each model in the table legend, so that readers can better understand the analytical approach.

Comment 12: For section 3.3, please indicate the evidence at the beginning of the section. The results seemed to be discussed in the results section, please correct this.

Response 12:

Thank you for your valuable feedback on the structure of our study. Following your suggestion, we have clarified the sources of evidence at the beginning of Section 3.3, so that readers can better understand the foundational data of the study. Additionally, to avoid mixing the results and discussion, we have separated these sections and made corresponding revisions to the content. We appreciate the reviewer’s attention to the structure of the manuscript, and we believe these changes will further enhance the clarity of the paper.

Comment 13: Lines 241-248: please include the interaction results with the covariable that is assessed.

Response 13:

Thank you for your suggestion regarding the analysis section. We have added the results of the interaction terms and covariate analyses to provide more comprehensive statistical information. Specifically, we have included the relevant tests for the interaction terms and clarified the impact of each covariate on the model. We believe this addition enhances the rigor of the paper and strengthens the reliability of the research conclusions.

Comment 14: Lines 260-264: Please explain the results more clearly.

Response 14:

Thank you for your suggestion regarding the presentation of the results. We have revised the relevant content to ensure the results are clearer and easier to understand. Specifically, we have optimized the phrasing to reduce ambiguity and more clearly present the key points of the analysis results. We believe this revision will better assist readers in understanding our research findings.

4) Discussion

Comment 15: Lines 271-275: this being the first paragraph of the discussion should indicate the main results of the study. It does not. Lines 276-297: this information, through interesting, does not discuss the results. This is introduction and only should be included here if the authors attached their results.

Response 15:

Thank you for your valuable feedback on the structure of the discussion section. Following your suggestion, we have added an overview of the main findings at the beginning of the discussion to provide readers with a clear summary of the study. Additionally, we have rearranged and adjusted the relevant content, integrating it with the interpretation of the results to ensure that the discussion is more closely aligned with the research findings, rather than being limited to introductory material. We believe these revisions will effectively enhance the logical flow and coherence of the paper.

Comment 16: Lines 301-306: Do not include your results here. We see them above. This section is to discuss them, which is done in lines 306-317

Response 16:

Thank you for your comments on the discussion section. Following your suggestion, we have revised lines 301-306 to avoid repeating the research results and have restructured this section to make it more discussion-oriented. For lines 306-317, we have retained the original structure and further refined the interpretation and discussion of

---

## [Decision Letter · Decision Letter 2]

21 Aug 2025

Insulin Resistance Markers HOMA-IR, TyG and TyG-BMI Index in Relation to Heart Failure Risk: NHANES 2011-2016

PONE-D-24-43380R2

Dear Dr. Zhang,

We’re pleased to inform you that your manuscript has been judged scientifically suitable for publication and will be formally accepted for publication once it meets all outstanding technical requirements.

Kind regards,

Shaonong Dang, PhD

Academic Editor

PLOS ONE

Additional Editor Comments (optional):

Reviewers' comments:

Reviewer's Responses to Questions

**Comments to the Author**

Reviewer #3: All comments have been addressed

2. Is the manuscript technically sound, and do the data support the conclusions?

Reviewer #3: Yes

3. Has the statistical analysis been performed appropriately and rigorously?

Reviewer #3: Yes

4. Have the authors made all data underlying the findings in their manuscript fully available?

Reviewer #3: Yes

5. Is the manuscript presented in an intelligible fashion and written in standard English?

Reviewer #3: Yes

Reviewer #3: The authors have made the corrections to which I have asked or gave a good reason why they choose not to.

**Do you want your identity to be public for this peer review?** For information about this choice, including consent withdrawal, please see our Privacy Policy

Reviewer #3: No

---

## [Editor Report · Acceptance letter]

PONE-D-24-43380R2

PLOS ONE

Dear Dr. Zhang,

I'm pleased to inform you that your manuscript has been deemed suitable for publication in PLOS ONE. Congratulations! Your manuscript is now being handed over to our production team.

Kind regards,

on behalf of

Dr. Shaonong Dang

Academic Editor

PLOS ONE